# Topics Mentioned by Stroke Patients during Supportive Phone Calls—Implications for Individualized Aftercare Programs

**DOI:** 10.3390/healthcare10122394

**Published:** 2022-11-29

**Authors:** Richard Schmidt, Daniela Geisler, Daniela Urban, Markus Wagner, Galina Ivanova, Thomas Neumuth, Joseph Classen, Dominik Michalski

**Affiliations:** 1Department of Neurology, University of Leipzig, 04103 Leipzig, Germany; 2German Stroke Foundation, 33311 Gütersloh, Germany; 3Innovation Center Computer Assisted Surgery (ICCAS), University of Leipzig, 04103 Leipzig, Germany

**Keywords:** stroke, TIA, stroke aftercare, topics in aftercare, aftercare concepts, individual need

## Abstract

By understanding stroke as a chronic disease, aftercare becomes increasingly important. For developing aftercare programs, the patients’ perspective regarding, for example, stroke-related symptoms and interactions with the healthcare system is necessary. Records from a local stroke pilot program were used to extract relevant topics from the patients’ perspective, as mentioned during a phone call two months after hospital discharge. Data from 157 patients with ischemic stroke or transient ischemic attack (TIA) were included. “Rehabilitation” was mentioned by 67.5% of patients, followed by “specialist physician”, “symptoms”, and “medication”. Compared with severely disabled patients, those with no relevant disability at hospital discharge mentioned “symptoms” significantly more often. Regarding rehabilitation, “outpatient care” was mentioned more often by patients in an inpatient setting, and 11.8% without rehabilitation mentioned “depression”. Patients in single-compared to multi-person households differed, for example, in the frequency of mentioning “specialist physicians” and gradually “outpatient care”. A multivariate model yielded associations between the disability at discharge and the probability of mentioning relevant topics afterward. This study provided insights into the patients’ perspective and identified topics that need attention while accompanying stroke and TIA patients after discharge. Further, the degree of disability at discharge might be helpful for planning individual aftercare.

## 1. Introduction

Stroke is characterized by the sudden onset of a focal neurological dysfunction due to infarction, caused by the occlusion of a cerebral vessel or hemorrhage, which is distinguished from transient ischemic attack (TIA) without structural damage [1]. Based on the subtype’s frequency, ischemic stroke is seen significantly more often (87%) than hemorrhagic stroke (13%) [2]. Recent data from the United States indicated that every year about 795,000 people experience a stroke, while a former evaluation of the situation in Germany yielded about 262,000 people per year [3]. In relation to the underlying populations, these numbers account for similar incidence rates, which are significantly lower than in low- and middle-income countries, where almost 90% of the stroke burden currently resides [4]. Overall, the number of strokes occurring each year is estimated at 7.6 million [5]. These huge numbers and the fact that stroke ranges among the first three causes of death worldwide qualify stroke as a relevant burden and a challenge to the healthcare system [5,6].

During the last three decades, enormous efforts have been made toward improved acute stroke treatment. This resulted in advances allowing now the re-opening of occluded cerebral vessels by either intravenous thrombolysis [7] or endovascular procedures [8] and multidisciplinary treatment in specialized centers, i.e., stroke units [6,9,10]. Although a reduction in the number of fatal courses was seen over the past years, a relative increase in the global lifetime risk of stroke became visible [6,10,11]. Further, in the first year after a stroke, about 11% of patients experience another cerebrovascular event, while 12 to 25% are affected by a secondary event within the following 5 years [1,12,13]. Regarding persisting neurological deficits, around 40% of patients are permanently impaired after stroke and thus depend on medical support or even institutional care [14]. By taking together the incidence of stroke and its consequences, a recent study estimated an increase of 27% in people living with stroke in the European Union up until 2047 [15]. Globally, there are more than 77 million people currently living who have experienced an ischemic stroke [5]. Estimates suggest that total global costs of stroke, including treatment, rehabilitation, social care, and informal caregiving, accounted for around 1.12% of the global gross domestic product [4,5].

Recently, stroke has been seen more and more as a chronic disease with a sudden onset and a variety of challenges in the long-term course after the event. This perspective strengthens the need for aftercare concepts that consider individual patient factors. The physician’s perspective on stroke aftercare traditionally involves a bundle of topics: Secondary prevention after an ischemic event is typically addressed by an individually adapted medication to interfere with the coagulation system, a modification of the lipid metabolism, treatment of arterial hypertension, lifestyle modifications for instance stop of nicotine consumption, or implementation of a healthy diet. Further, the modification of already existing risk factors is typically accompanied by the search for hardly detectable factors such as paroxysmal atrial fibrillation [16]. Aftercare also includes the best possible reduction in neurological deficits that remain from the cerebral event, which provides for rehabilitative actions in rehabilitation centers or the ambulatory setting, and often long-term physical therapy [17]. A further part of aftercare is the detection and treatment of secondary diseases, for instance, depression, which is known to occur in up to 29% of patients [18].

However, existing aftercare frameworks often make it difficult for patients, carers, and involved health professionals to establish a well-coordinated, individually adapted process [10,17,19]. The prevailing problems include, for example, a lack of availability of relevant medical information at different interfaces, and the sectoral separation of care, making it difficult for patients to use services from various healthcare professionals in the chronic phase following acute stroke. These limitations are illustrated by a German study, which is based on a survey among experienced physicians in the field of stroke aftercare, and yielded that patients affected by a stroke often must organize their appointments with specialist physicians and therapists in addition to their general practitioner [20]. In addition, case-related professionals in the post-hospital setting, who would be able to guide patients and their relatives, are usually lacking. Consequently, many patients who experience a stroke or TIA perceive hurdles that prevent them from effectively discussing their needs and accessing services [21]. Because of these difficulties, stroke pilots were discussed as a personalized element of the aftercare landscape to facilitate the transition from the hospital to the post-hospital setting, to give knowledge on stroke, to accompany structured treatment pathways, and to provide individual help in the long-term course after the event [22]. Stroke pilots are healthcare professionals who subsequently specialize in case management and aftercare. While most stroke pilots are likely to have their training background in nursing, they could have also been physio- or speech therapists before working in the field of stroke aftercare. As one of their activities is helping patients navigate through the healthcare system, they might also be called stroke guides, stroke navigators, or stroke nurses. While national and international efforts are made to unify the training and qualification of professionals specializing in stroke aftercare, the effects of stroke pilots are currently under evaluation. However, the first data from a German and Danish randomized trial, including stroke and TIA patients, indicate that a structured aftercare program compared to usual care may result in improved control of risk factors, even though the rate of vascular events was not reduced during a follow-up of 3.6 years [23]. Further, an Austrian randomized trial indicated a better quality of life and a decreased rate of major cardiovascular events by a structured aftercare program compared to usual care [24].

Along with the increasing awareness of stroke aftercare with its individual and structural challenges, initiatives such as the Stroke Action Plan for Europe were started recently by the European Stroke Organisation (ESO) and patient organization (SAFE) that highlight, among others, the patients’ perspective while focusing on the patients’ outcome and individual conditions in daily living [25]. Consequently, research has begun to explore the patients’ individual experiences through patient-reported outcome measures, including, for instance, quality of life and physical fitness [26,27]. Thereby, the fact that patients themselves evaluate a situation that has arisen, for example, along with medical intervention, is seen as an essential step towards establishing patient-oriented health services. In addition to the discussion regarding the kind of evaluation, knowledge of crucial contents is important when developing patient-centered aftercare programs. For this purpose, topics relevant from the patients’ perspective regarding daily living, stroke-related symptoms, and interactions with the healthcare system have a pivotal role.

This study aimed to identify existing topics in stroke aftercare from the patients’ perspective while using data from a demand-oriented and thus mainly unstructured stroke pilot program in a real-life setting starting shortly after an ischemic stroke or TIA. Factors that may affect the patients’ perspective, physical disability, the type of rehabilitation, and the social environment were considered.

## 2. Materials and Methods

### 2.1. Study Design

By applying a retrospective design, this study used data recorded during a local stroke pilot program. Patients with ischemic stroke or TIA treated at the Stroke Unit of the Department of Neurology at the University of Leipzig between January 2020 and January 2022 were offered support by stroke pilots. The support included individualized oral and written information related to the qualifying cerebrovascular event and was given during the hospital stay and later if necessary. After hospital discharge, patients had the opportunity to call the stroke pilots for further information or individual help regarding any content related to the qualifying event. Patients were informed that stroke pilots would contact them two months after hospital discharge for an unstructured interview addressing the current situation and individual problems and providing further help. The two stroke pilots were certified nurses, experienced in clinical stroke care, trained in care and case management (German Society for Care and Case Management), and received education in stroke aftercare (German Stroke Foundation) during the program.

This study was conducted in accordance with the guidelines of the Declaration of Helsinki and was approved by the institutional review board of the Medical Faculty of the University of Leipzig (reference number 081/22-ek). Further, the study was registered in the German Clinical Trial Register (reference number DRKS00030540).

### 2.2. Data Extraction and Processing

From each patient who had participated in the local stroke pilot program, clinical (e.g., National Institutes of Health stroke Scale (NIHSS; [28]), modified Rankin scale (mRS; [29])) and socio-demographic data assessed during the hospital stay were extracted from medical records.

Two months after discharge, stroke pilots contacted patients by phone to lead unstructured interviews while trying to document all relevant content of the conversation. In order to extract key topics of these phone conversations, the paper-based transcripts of the unstructured interviews during phone calls between pilots and patients were analyzed. Keywords mentioned by patients during phone calls were identified from the transcripts and digitally listed for each patient. During this process, multiple different keywords for one patient could be recorded, while duplication of one keyword at the level of one patient was not allowed. Four persons (two stroke pilots and two physicians) re-assessed a list of all keywords assigned to patients and merged keywords covering overlapping topics if necessary while still trying to keep relevant fields specific and distinguishable, resulting in a list of possibly multiple keywords assigned to each respective patient. During the following analyses, keywords were used as representative topics.

### 2.3. Statistical Analyses

The study population’s clinical, socio-demographic, and rehabilitation-related characteristics were analyzed descriptively. For subgroup analyses, patients were categorized according to clinically and socio-demographically relevant factors. Factors represent the degree of disability or dependence in daily activities at hospital discharge (assessed by the mRS), the type of rehabilitation after hospital discharge, and the patients’ social environment (depicted by the number of persons living in the patients’ household).

Topics mentioned in the overall study population or subgroup of patients were analyzed according to their frequencies, respectively, while for graphical illustration, the 10 most often mentioned topics were mapped. Pearson’s Chi-squared test of independence with Yates’ continuity correction (if applicable) was used to evaluate the association between mentioning a topic and belonging to a specific clinical or socio-demographic group. The influence of clinical or socio-demographic features at the time of hospital stay on the odds of mentioning a particular topic in the phone call two months later was assessed with multivariate logistic regression, and adjusted odds ratios were determined for features considered clinically or socio-demographically relevant factors.

The significance level for all tests was 0.05, and a false discovery rate < 0.25 was adjusted using the Benjamini–Hochberg method. Analyses were performed with the software R version 4.2.1 [30] with R Studio [31].

## 3. Results

### 3.1. Patients’ Characteristics and Frequency of Topics Mentioned after Hospital Stay

Of 175 patients who participated in the stroke pilot program, 157 patients (89.7%) could be reached by phone two months after discharge from the hospital and were included in the present study. Of these patients, three had a TIA (1.91%), while the majority of patients were diagnosed with an ischemic stroke as the qualifying event, characterized by a wide range of clinical affection at hospital admission (NIHSS ranging from 1 to 22) and different degrees of disability at hospital discharge (mRS ranging from 1 to 4) (Table 1). Remarkably, most patients attended any type of rehabilitation after their hospital discharge, while about 12% of patients did not attend. The main reasons for a missed rehabilitation were that patients refused to participate in a rehabilitation program or that pension insurance denied rehabilitation reimbursement. Most patients shared their household with at least one further person (e.g., partner or parent) after hospital discharge.

Overall, the most frequent topic in the study population was “rehabilitation”, as it was mentioned by nearly two-thirds of all patients, followed by medical topics such as “specialist physician”, “symptoms”, and “medication” (Figure 1). Less frequently mentioned topics in the overall study population dealt with stroke-related consequences regarding occupation and independent mobility. The comparatively large but heterogeneous list of topics mentioned least frequently, often mentioned by only one or two patients, included “psychotherapy”, “concentration”, “fatigue”, “anxiety”, and “memory”, as well as “smoking”, “sleep apnea”, “self-help group”, and “disease awareness” (Appendix A).

### 3.2. Frequency of Mentioned Topics Depending on Disability and Social Environment

Distributions of topics were analyzed regarding subgroups considering the degree of disability, type of rehabilitation after hospital discharge, and the social environment. Around twice as many patients left the hospital with a relevant impairment in daily living (mRS > 1) compared to those with no or a non-relevant disability (mRS 0–1) (Figure 2). While “rehabilitation” was mentioned in a comparable frequency, patients with low mRS at hospital discharge mentioned “symptoms” significantly more often (*p* = 0.0001) during the phone call two months later. Patients with mRS > 1 mentioned “aids and appliances” (*p* = 0.0026), “outpatient care” (*p* = 0.0164), and “disability” (*p* = 0.0124) significantly more often than patients with mRS 0–1 at hospital discharge. Remarkably, topics such as “specialist physician”, “medication”, and “blood pressure” were mentioned gradually more often by patients with mRS > 1; however, it did not reach statistical significance. Expectedly, topics such as “sports” and “occupation” were mentioned gradually more often by patients with no or a non-relevant disability (mRS 0–1) at the time of hospital discharge.

Concerning rehabilitation, most patients had attended some form of a rehabilitation program after discharge from the hospital in either an inpatient or outpatient setting (Figure 3). When grouped for the type of rehabilitation, the topic “outpatient care” was found to have the highest frequency in the group that had received inpatient rehabilitation, with a significant difference between all three groups (*p* = 0.0100). It was noticeable that topics such as “aids and appliances”, “disability”, and “therapy” were mentioned gradually more often in the inpatient rehabilitation group compared to the group covering outpatient rehabilitation; however, it did not reach statistical significance. Expectedly, the topics “sports” and “occupation” were gradually more often mentioned in the outpatient rehabilitation group. Remarkably, the topics “rehabilitation”, “symptoms”, and “special physician” represented the three most mentioned topics in the groups of patients receiving outpatient and inpatient rehabilitation, while patients with no rehabilitation mentioned “rehabilitation”, “specialist physician”, and “medication” as the three most mentioned topics, followed by “driving license”. However, with 11.8% in the group that had received no rehabilitation, “depression” was mentioned twice as often as in other groups, even though a statistically significant difference among the groups did not exist.

Regarding the patients’ social environment, the number of persons living together with a patient in one household was considered for subsequent analyses while stratifying into a single-person and a multi-person household (Figure 4). Thereby, topics such as “outpatient care”, “home environment”, “aids and appliances”, and “specialist physician” were mentioned gradually more often by patients living in a single-person household than those living in a multi-person household, even though no statistical significance could be detected. The list of most frequently mentioned topics in multi-person households was mainly reflected by medicine-related topics such as “specialist physician”, “symptoms”, “blood pressure”, and “medication”, while “specialist physician” and “blood pressure” were significantly more often mentioned in this group when compared to patients in a single-person household (*p* = 0.0377; *p* = 0.0490).

### 3.3. Probability for Mentioning Topics in Stroke Aftercare Depending on Patients’ Characteristics and Social Environment

The impact of clinical and socio-demographic features such as disability at the time of discharge, type of rehabilitation promptly after discharge, and the number of persons living together with the patient on the probability of having mentioned one of the most frequent topics were assessed with a multivariate logistic regression and adjusted odds ratios. Iteratively for each of the most frequently mentioned topics, a model was fitted with mentioning the topic versus not mentioning the topic as the dependent dichotomous variable in the logistic regression. Age, sex, NIHSS at admission, number of secondary diagnoses per patient, number of cardiovascular risk factors per patient, days spent in hospital, NIHSS and mRS at the time of hospital discharge, number of prescribed medications at the time of discharge, type of rehabilitation after discharge, and number of persons living in patients’ household were included in each respective model as explaining variables (detailed results in Appendix A).

Concerning the patients’ age, the probability of mentioning the topics “outpatient care”, “aids and appliances”, and “home environment” doubled every ten years, while the odds of having mentioned “occupation” decreased to a third over the same period of years of age. Remarkably, women were less likely to mention “medication” but around six times as likely as men to have mentioned “outpatient care”. Regarding the severity of neurological symptoms at hospital admission as measured by the NIHSS, the probability of having mentioned “home environment” two months later increased by 18% with every point in NIHSS at the time of admission.

While adjusting for the above-mentioned socio-demographic and clinical characteristics, a higher disability (mRS) at hospital discharge significantly increased the probability of a patient having mentioned the topics “aids and appliances”, “disability”, and “driving license”, respectively, in the follow-up two months after hospital discharge (Figure 5). With every score increase in mRS at discharge, patients were around five times as likely to have mentioned the topic “aids and appliances” and about three times as likely to have mentioned “driving license” or “disability”. The odds of having mentioned the topics “rehabilitation” and “symptoms” was nearly halved by every score increase in mRS. Although the type of rehabilitation did not seem to have any significant influence on the odds of having mentioned any of the most frequent topics, the odds of having mentioned “blood pressure” were more than four times as high in the group of patients living together with at least one other person compared to patients living in a single-person household.

## 4. Discussion

This study was focused on stroke aftercare, which becomes increasingly important because the management of patients’ varying disabilities and individual problems in daily living is still challenging, while substantial progress concerning acute treatment was already achieved. Considering current initiatives such as the Stroke Action Plan for Europe and SAFE [25], this study, in particular, addressed the patients’ perspective during the aftercare and thus aimed to identify relevant topics for people who experienced an ischemic stroke or TIA.

Observations were based on a local stroke pilot program that starts during the hospital stay and includes a phone call two months after discharge from the hospital, where patients had been treated for the qualifying cerebrovascular event. The implementation of stroke pilots was driven by a previous initiative suggesting overall beneficial effects in stroke aftercare due to such pilots [22] and studies showing advantages, for example, regarding compliance [32], quality of life [24], and emotional well-being of patients in nurse-supported aftercare programs [33]. In addition to topics mentioned during the follow-up, individual clinical and socio-demographic features assessed during the hospital stay and the type of rehabilitation patients attended after discharge were considered.

In the overall study population, patients most frequently mentioned the topic “rehabilitation”, which is likely related to the fact that most patients have already attended or still attended a rehabilitation program in an inpatient or outpatient setting at the time of the phone call. Although a rehabilitation program after a stroke is reasonable in most stroke cases from a medical point of view, as not only the reduction in neurological deficits is relevant, but risk factor control and lifestyle modifications also represent individual issues, access to rehabilitation seems to be different among countries. While in the present study, the high rate of people attending inpatient rehabilitation (about 58%) might indicate easy access in Germany, a review regarding regional differences described only a rate of 13% in Sweden and 57% in Israel [34]. However, such observations, which are probably related to different orientations of healthcare systems, might be considered while prioritizing elements for regional aftercare programs.

Frequent topics were also “specialist physician” and “medication”, representing typical elements of stroke aftercare. This finding might indicate that patients are aware of these topics, which could be seen as a success since these elements became part of the patients’ knowledge. The hypothesis that patients are particularly interested in their health might be supported by an observation made in another study looking at stroke patients supported by a message system, where most of them chose their health as a significant goal domain to be addressed in stroke aftercare [35]. On the other hand, the results of this study could also suggest that problems in these areas arose during aftercare. Problems may include appointments with specialist physicians and therapists, as identified as an issue in an earlier report [20]. Logistical, technical, and hidden barriers such as missing consistent referral processes, appointment wait time, insufficient knowledge about local health care services, or even fear of medical care could have impacted access to specialist physicians after hospital discharge [36,37]. As earlier studies have shown, medication adherence is low in patients who experienced a stroke which could possibly affect the frequency of mentioning the topic in both directions [38]. Whenever possible, future aftercare programs should therefore acknowledge barriers experienced by patients in order to optimize their personal long-term care, including, for instance, medication adherence as well as risk factor monitoring.

In the present study, about one-third of patients mentioned “symptoms” two months after discharge, indicating that a relevant proportion realized some kind of symptoms after that period, which appears relatively high when considering the low NIHSS and mRS, each with a mean of about 1.7, at the time of hospital discharge. On the other hand, persisting problems in patients with low or moderate deficits due to stroke were also described in other studies: For example, in a cohort of stroke patients with a median mRS of 2 at hospital discharge, a proportion of about 24% had an mRS of more than 2 after 12 months [24]. These findings strengthen the perspective that clinical symptoms need to be re-assessed also in the long-term course of a stroke, thus qualifying respective screenings as essential elements in stroke aftercare programs as a prerequisite for sufficient detection and initiation of treatment approaches.

By considering the degree of disability at hospital discharge, the type of rehabilitation, the social environment, and socio-demographic characteristics, this study suggested that these factors may impact the topics relevant from the patients’ perspective two months after stroke. Unexpectedly, the topic “symptoms” was mentioned significantly more often by patients with no or a non-relevant disability at hospital discharge than those with a more significant disability. One hypothesis might be that phenomena such as fatigue or impaired memory function, usually not seen as typical stroke-related symptoms such as hemiparesis or aphasia, have become relevant for daily living. Observations of the present study are in line with earlier findings, showing that the degree of disability and dependence on daily activities at the time of discharge are statistically related to long-term disability after stroke [39,40]. In detail, patients being relatively more disabled at hospital admission significantly more often mentioned “disability” after two months, which might be interpreted as an awareness of existing physical impairments by the patients themselves. Further, more disabled patients at hospital admission mentioned “aids and appliances” and “outpatient care” during the further course, which might show the existing need for exceptional support from the healthcare system. When focusing on aspects associated with social life, patients with no or a non-relevant disability at hospital discharge mentioned topics such as “sports” or “occupation” with a gradually higher frequency than patients with higher disability, indicating that those with lower disability sought access to or already participated in social life at a higher degree. When focusing on the type of rehabilitation and social environment, significant differences in the frequency of topics mentioned two months after hospital discharge was found concerning “outpatient care”, which was highest in the group of patients attending rehabilitation in an inpatient setting. One explanation might be that patients receiving intensive rehabilitation in an inpatient setting become aware of the meaningfulness of continuing rehabilitative elements in the following outpatient setting. Concerning secondary diseases such as depression, the present study indicates that particularly in patients receiving no rehabilitation, depression represents a relevant topic as this was mentioned twice as often when compared to patients who attended inpatient or outpatient rehabilitation. Consequently, when designing aftercare programs for patients with cerebrovascular events, implementing a systematic screening approach, especially in patients not medically accompanied in a structured path for diagnostics and treatments, as usually performed during rehabilitation, seems necessary. Such an approach also appears reasonable in patients suffering from a TIA or minor stroke with, per definition, no or minor neurological deficits since earlier observations have indicated that about 20% of these patients exhibit signs of depression when using an established scoring system [41]. While focusing on the social environment, this study identified the topics “specialist physician” and “blood pressure” more often mentioned by patients living in multi-person households than those in single-person families. While “symptoms” was mentioned with a comparable frequency of at least about 25%, “outpatient care” and “home environment” was gradually more often mentioned by patients living in single-person households than those in multi-person households. This observation seems reasonable as both groups are likely aware of existing symptoms, but their management is supported by other persons than the patients in multi-person households. Regarding socio-demographic factors, the findings of the present study are in good accordance with earlier investigations on sex differences in strokes, where women were much more likely to demand “outpatient care” but less likely to mention “medication” [42].

While applying a multivariate logistic regression model controlled for socio-demographic, clinical, and other factors, individual disability at hospital discharge seemed to provide a robust parameter that is statistically associated with the probability of mentioning topics such as disability, “aids and appliances”, and “driving license” during the period after hospital discharge.

This study has some limitations: First, the retrospective design and the necessary data extraction from unstructured notes might have impacted the range and number of topics that became assessable for the analyses. Even though reviewers tried to merge overlapping topics and keep relevant fields distinguishable, opinions and topics could have been missed in the analysis because they might have been judged not specific enough. Moreover, quantitative analysis of topics did not account for a topic being mentioned multiple times by one patient during one phone call because the underlying, i.e., handwritten, transcripts were not reliable enough for such an analysis. While taking notes, stroke pilots could have missed specific words or phrases mentioned by patients, which then did not go into our analyses. Further, due to the necessary grouping of conversation topics, the direction of the association could be challenged. For example, it was not definite that mentioning “aids and appliances” implied demand in this area or was mentioned because no more need was present. Second, stroke pilots could have been biased in their conversation to generate topics depending on medical information from discharge. Moreover, certain patients’ topics could have been missed because conversations were led by phone and not present, enabling patients to conceal specific problems. If stroke pilots had visited patients at home, other requirements that patients did not actively mention by themselves could have been more obvious. In addition, further topics might have been generated by investigating the caregiver’s or professional’s opinion on the patients’ needs, which has been performed in earlier approaches [43,44]. Third, because of the monocentric and, in proper meaning, non-interventional study design, the sample size is relatively small, and a control group is lacking, which limits the generalizability of the study findings. The sample size becomes relevant, especially in the performed regression model, while the number of patients with higher degrees of disability may have been too small to reach consistent power. Further, more patients with TIA would have allowed a stratification against ischemic stroke, which was impossible in the available data set. Fourth, the observation period used in this study is relatively short. Due to the fact the individual courses after stroke and TIA vary in a wide range with implications regarding individual needs, topics relevant from the patients’ perspective may naturally change over time. By considering these limitations, the present study’s findings need to be confirmed in larger cohorts of patients with cerebrovascular events and with a longer follow-up.

However, to the best of our knowledge, this is one of the first studies that identified topics during stroke aftercare directly from patients having experienced ischemic stroke or TIA and thus provides valuable insights into the patients’ perspective. Therefore, these insights can be used to improve existing or generate new initiatives in stroke aftercare, which is naturally individual and thus highly complex. In detail, persisting symptoms due to stroke, access to specialist physicians, and handling of medication or knowledge on it, together with the broad field of rehabilitative aspects, seem to represent topics that need further attention by persons involved in stroke aftercare or perhaps multi-professional programs accompanying stroke and TIA patients after hospital discharge. Furthermore, among factors that were easily assessable during the hospital stay, the degree of disability seems to represent a valuable parameter for adapting aftercare programs in a patient-centered way, as in the present study, this factor was identified to have a robust statistical relationship to topics that have become relevant during the following two months. Moreover, the type of rehabilitation and the social environment appear suitable for planning individual aftercare concepts and may help to guide post-discharge conversations between healthcare professionals and patients.

## Figures and Tables

**Figure 1 healthcare-10-02394-f001:**
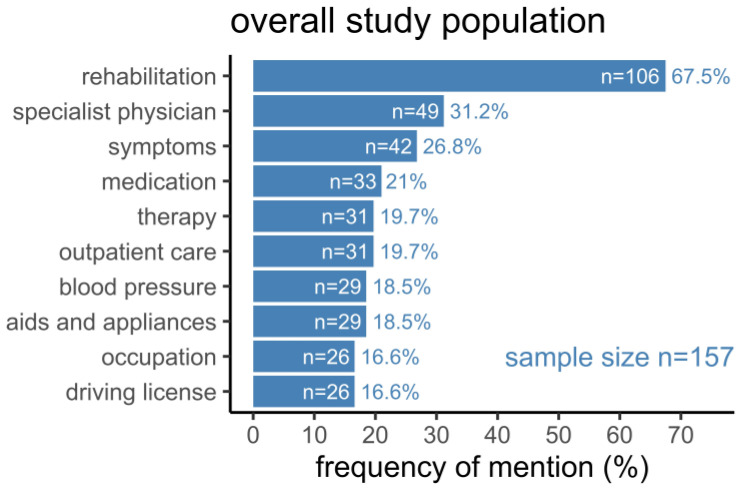
Topics mentioned during a stroke pilot-based phone call two months after hospital discharge. Bars represent the number of patients (n) who mentioned a specific topic. Added rates represent the percentage of patients (%) with respect to the underlying sample size.

**Figure 2 healthcare-10-02394-f002:**
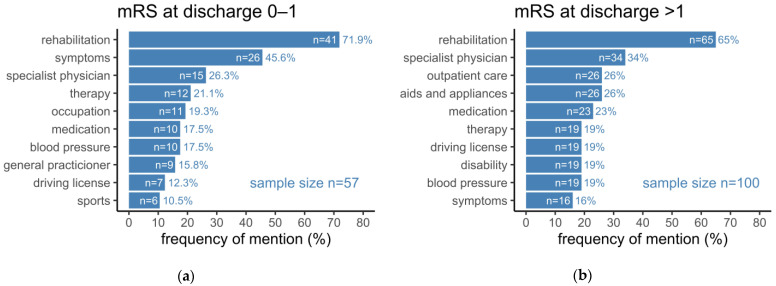
(**a**) Topics mentioned by patients with no or a non-relevant disability (modified Rankin scale (mRS) 0–1) at discharge from hospital; (**b**) Topics mentioned by patients with more severe disability (mRS > 1) at discharge from hospital. Bars represent the number of patients (n) who mentioned a specific topic. Added rates represent the percentage of patients (%) with respect to the underlying sample size.

**Figure 3 healthcare-10-02394-f003:**
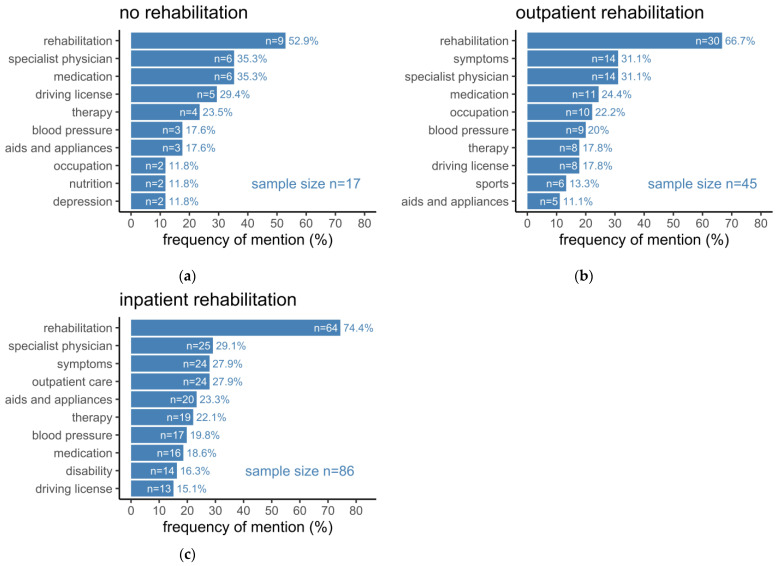
(**a**) Topics mentioned by patients who have not attended rehabilitation; (**b**) Topics mentioned by patients who have attended outpatient rehabilitation; (**c**) Topics mentioned by patients who have attended inpatient rehabilitation. Bars represent the number of patients (n) who mentioned a specific topic. Added rates represent the percentage of patients (%) with respect to the underlying sample size.

**Figure 4 healthcare-10-02394-f004:**
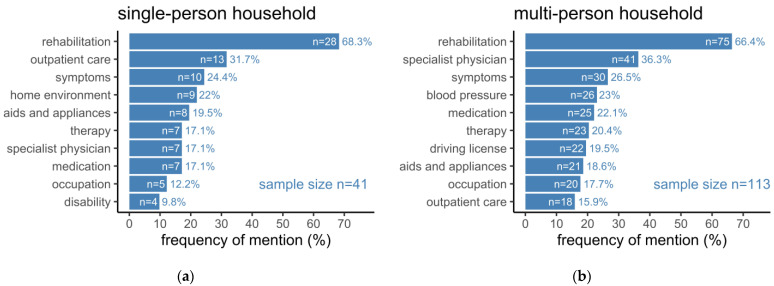
(**a**) Topics mentioned by patients living in a single-person household; (**b**) Topics mentioned by patients living in a household with at least one more person. Bars represent the number of patients (n) who mentioned a specific topic. Added rates represent the percentage of patients (%) with respect to the underlying sample size.

**Figure 5 healthcare-10-02394-f005:**
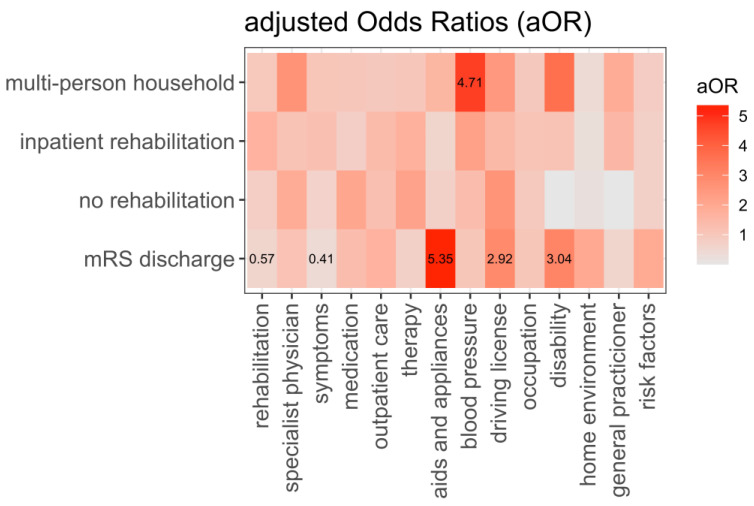
Statistical relationship between clinical and socio-demographic variables during the hospital stay and mentioned topics two months after hospital discharge. Heat map shows respective adjusted odds ratios (aOR) for mentioning a specific topic while considering age, sex, NIHSS at admission, number of secondary diagnoses, number of cardiovascular risk factors, days spent in hospital, NIHSS at time of discharge, and the number of prescribed medications at time of discharge. aOR were calculated with a logistic regression model with mentioning versus not mentioning the respective conversation topic as a dependent dichotomous variable (*x*-axis) and clinical and socio-demographic factors as explaining variables (*y*-axis) adjusted for other clinical and socio-demographic variables. Fields with significant results for aOR are labeled with the exact aOR.

**Table 1 healthcare-10-02394-t001:** Baseline characteristics of patients contacted by phone.

Characteristic	M ± SD or n (%)
Age (years)	69.32 ± 13.21
Sex (female)	57 (36.31%)
BMI	27.45 (4.97%)
Diagnosis	
TIA	3 (1.91%)
Ischemic stroke	154 (98.09%)
Cardiovascular risk factors per patient	3.74 ± 1.73
Recanalization strategy	
None	115 (73.25%)
Intravenous thrombolysis	34 (21.66%)
Endovascular procedure	24 (15.29%)
Patients with first stroke	129 (82.69%)
Secondary diagnoses per patient	6.16 ± 3.23
pre-mRS	
total	0.29 ± 0.68
0	127 (80.89%)
1	18 (11.46%)
≥2	12 (7.64%)
NIHSS at hospital admission	
total	4.09 ± 4.32
0	25 (15.92%)
1–3	67 (42.68%)
4–8	43 (27.39%)
≥9	22 (14.01%)
Days in hospital	8.82 ± 5.14
mRS at hospital discharge	
total	1.77 ± 1.10
0	23 (14.65%)
1	34 (21.66%)
2	66 (42.04%)
≥3	34 (21.66%)
NIHSS at hospital discharge	
total	1.69 ± 2.27
0	55 (35.03%)
1	43 (27.39%)
2–3	42 (26.75%)
≥4	17 (10.83%)
Medications ^1^ at hospital discharge	6.64 ± 2.89
Type of rehabilitation ^2^	
none	17 (11.49%)
outpatient	45 (30.41%)
inpatient	86 (58.11%)
Patients in single-person households ^2^	41 (26.62%)

M: mean; SD: standard deviation; BMI: body mass index; TIA: transient ischemic attack; mRS: modified Rankin Scale (ranging from 0 (no symptoms at all) to 5 (severe disability requiring constant care) [29]); pre-mRS: mRS before hospital admission; NIHSS: National Institute of Health Stroke Scale (ranging from 0 to 42 indicating specific fields of impairment [28]); ^1^ including any type of medication, for instance, anticoagulant, antihypertensive, or diabetes medication; ^2^ non-available data not listed.

## Data Availability

To preserve the anonymity of study participants, the underlying data could not be made available, and data-sharing, for instance, for meta-analyses, was not contemplated in the study protocol.

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
