# Peer review of "Topics Mentioned by Stroke Patients during Supportive Phone Calls—Implications for Individualized Aftercare Programs"

_healthcare, 2022, doi:10.3390/healthcare10122394_

Round 1
Reviewer 1 Report
In the background, a clearer division should be made between international and national developments. In addition, epidemiological figures on stroke care should be added, if available. Likewise, the advantages and disadvantages of patient-oriented health services research should be addressed.
The methods section should be significantly revised. On this basis, it is not possible to gain a clear idea of how exactly the data were collected and documented and in what quality they are available or in what way quality management was carried out. This raises the question of whether the very broad categories could be made more specific. Also the selection process of the categories seems to have been based mainly on quantitative criteria. This should be reflected critically. How was multiple responses handled in the analysis?
The presentation of the multivariate analysis in the heat map is in interesting but difficult to read.
The discussion is essentially a presentation of the results; here a stronger reference to the healthcare situation, e.g. what are indications of individual and or structural problems of care, as well as to patient-oriented health services research would be desirable.
Reviewer 2 Report
Thank you for the opportunity to review this interesting paper. The information in the paper may be helpful to guide post-discharge conversations.
I think there are a number of aspects that require addressing before this paper is published (summarised below)
Each occurrence of 'suffers from' should be changed to 'experienced' or terminology that is less pejorative (pg 1 ln 34, and 44, pg 4 ln 154)
Ln 33-35 discusses the number of people experiencing stroke in US and Germany, it would provide a better comparison if expressed in relation to size of population.
Avoid using 'so-called' and define terms instead.
Pg2 ln 49 - I think 'until' is incorrect word-consider reviewing.
It would be helpful to define stroke pilot in Introduction. The final paragraph of the Introduction really just needs to state aim, rather than moving into aspects of Method.
In the Method, you indicate that the interviews were unstructured, it would be helpful to describe how the interviews were initiated. Were there any starter questions? How were these undertaken?
Also, what was documented from these interviews. Did the records attempt to document all topics, or just those that need actions. This information would help understand how comprehensive the recorded responses are likely to be.
There seemed to be some confusion about the timings of the clinical measures (at discharge/"at time of hospital stay" and the interviews 2 months after discharge/2 months after event). These need clarifying throughout paper.
Consider use of term Therapy eg in Table 1, but also in text in Results 207-just clarify if physio/OT etc or medical intervention.
Pg4 ln157 needs re-wording
Figure 1 does not include all topics-which were included, how were they chosen?
pg6 ln 193-failing statistical significance, consider instead re-phrasing to 'does not reach statistical significance' or similar.
Final line of paper is maybe a little misleading as I don't think the message is that depression should be detected based on the findings from this study.
